# PYRAMID RECURRENT NEURAL NETWORKS FOR MULTI-SCALE CHANGE-POINT DETECTION

## ABSTRACT

Many real-world time series, such as in activity recognition, finance, or climate science, have changepoints where the system's structure or parameters change. Detecting changes is important as they may indicate critical events. However, existing methods for changepoint detection face challenges when (1) the patterns of change cannot be modeled using simple and predefined metrics, and (2) changes can occur gradually, at multiple time-scales. To address this, we show how changepoint detection can be treated as a supervised learning problem, and propose a new deep neural network architecture that can efficiently identify both abrupt and gradual changes at multiple scales. Our proposed method, pyramid recurrent neural network (PRNN), is designed to be scale-invariant, by incorporating wavelets and pyramid analysis techniques from multi-scale signal processing. Through experiments on synthetic and real-world datasets, we show that PRNN can detect abrupt and gradual changes with higher accuracy than the state of the art and can extrapolate to detect changepoints at novel timescales that have not been seen in training.

## 1 INTRODUCTION

Changepoints, when the structure or parameters of a system change, are critical to detect in many domains. In medicine, finance, climate science and other fields, these changes can indicate that important events have occurred (e.g. onset of illness or a financial crisis), or changed in important ways (e.g. increasing illness severity). In both cases, these affect decision-making. Changepoint detection (CPD) aims to find these critical times. However, changes may result in complex patterns across multiple observed variables, and can be hard to recognize, especially in multivariate time-series where interdependencies exist among variables. Further, not all changepoints lead to a sudden transition, many occur over a duration of time (e.g. weightloss, transition between activities) and are harder to identify.

Various methods have been proposed for CPD including parametric methods (Adams & MacKay, 2007; Zhang et al., 2010; Montanez et al., 2015), which make strong assumptions about data distributions, and nonparametric methods (Desobry et al., 2005; Saatçi et al., 2010; Li et al., 2015), which are based on engineered divergence metrics or kernel functions. Most parametric methods are highly context specific, and face difficulty when changes result in complex temporal patterns that are hard to model manually. For nonparametric methods, the main drawback is that these methods rely heavily on the choice of parameters or kernels. To handle data from different domains, Chen et al. (2015) proposed a nonparametric CPD method. However, like many other CPD methods, it can only detect abrupt changes. Yet in real-world applications, the effect of a change may be gradual and may happen over different durations. Some methods have been explicitly designed for detecting gradual changepoints (Bardwell & Fearnhead, 2017; Harel et al., 2014), but cannot handle changes occurring at arbitrary timescales. In some applications, like detecting changes in activity, how quickly someone transitions from sitting to standing should not affect accuracy at detecting the transition.

In contrast, Deep Neural Networks (DNN) have been used for time series forecasting (Weigend, 2018) and classification (Yang et al., 2015) as they can learn functions automatically. These can be more easily adapted to new tasks if there is sufficient training data. However, DNNs typically need enough examples of all possible ways a pattern can appear, and thus all possible transition speeds,

to reliably detect it in test data. Since this data is costly and may be infeasible to collect in some cases, it is ideal to have a scale-invariant approach that can generalize beyond observed timescales.

We propose a novel DNN architecture for CPD using supervised learning. Our approach makes two key contributions to neural network architecture: a trainable wavelet layer that transforms input into a pyramid of multiscale feature maps; and Pyramid recurrent neural networks (PRNN), which build a multi-scale Recurrent Neural Network (RNN) on top of a multi-channel Convolutional Neural Network (CNN) processing the wavelet layer. Finally, we use a binary classifier on the PRNN output to detect changepoints. On both simulated and real-world data, we show that the proposed model can encode short-term and long-term temporal patterns and detect from abrupt to extremely gradual changepoints. The model is scale invariant, and can detect changes at any timescale, regardless of those seen in training. We focus on the task of CPD, but this architecture may have more general applications in time series analysis.

## 2 RELATED WORK

**Changepoint detection** CPD is a core problem for times-series analysis. One approach is to use a model and find times when observations deviate from what is predicted by the model. Bayesian Online ChangePoint Detection (BOCPD) (Adams & MacKay, 2007) can find changepoints in an online manner, but makes the limiting assumption that the time series between changes has a stationary exponential-family distribution. More generally, Bayesian techniques require full definition of the likelihood function (Malladi et al., 2013; Montanez et al., 2015), which may be difficult to specify. Nonparametric models increase the flexibility, such as in (Saatçi et al., 2010) which is an extension of BOCPD to Gaussian Processes. This however, may significantly increase computational complexity. Xuan & Murphy (2007) introduced Gaussian Graphical Models (GGMs) for CPD, extending (Fearnhead, 2006) to handle mutlivariate time series. GGM is offline and models the correlations between multivariate time series using multivariate Gaussian. This method is closest to ours as a result, but makes strong assumptions about the data distribution. Non-Bayesian techniques exist, such as (Yamanishi & Takeuchi, 2002), which uses an autoregressive model for each time series segment, but this model is limiting.

To eliminate the need to specify a model, model-free approaches have emerged, such as density-ratio estimation methods (Yamada et al., 2013; Kawahara & Sugiyama, 2012; Liu et al., 2013; Kuncheva, 2013; Kuncheva & Faithfull, 2014), kernel methods (Harchaoui et al., 2009; Li et al., 2015), and other techniques that define custom divergence functions like difference of covariance matrix (Cabrieto et al., 2017; Barnett & Onnela, 2016) or carefully engineered statistics (Cavalcante et al., 2016; Idé et al., 2016; Qahtan et al., 2015; Li et al., 2015; Gretton et al., 2007). However, covariance matrix based methods cannot deal with the case when the change point does not cause significant variations in covariance matrix. Statistics based methods such as MMD (Gretton et al., 2007), Hotellig T-square (Chen & Gupta, 2000), CUSUM (Page, 1954), or generalized likelihood ratio (GLR) (James et al., 1992) have their own limitations like relying heavily on the choice of kernels (MMD) or parameters (Hotellig T-square), being highly dependent on prior information (CUSUM), or having high complexity for large sample size (GLR). Thus while such models might work in a specific application, they cannot be readily used in a different domain without re-engineering the divergence or kernel functions.

Few methods were explicitly designed to detect gradual changes, though BOCPD has been extended this way by reformulating changes as segments instead of points (Bardwell & Fearnhead, 2017). Alternatively, gradual changes can be formulated as concept drifts (Harel et al., 2014). We do not reformulate the changepoint detection problem, and instead make the model scale-invariant, so it can handle short- and long-term temporal patterns. This results in a model that can generalize to novel time-scales without extra effort.

A similar problem is anomaly detection (Jones et al., 2016; Guha et al., 2016). For instance, Gardner et al. (2006) learns a one-class Support Vector Machine (SVM) on normal data, and distinguishes normal from abnormal in new data. However, a changepoint is not always a transition to an abnormal state and may be between two normal states, such as human activities. Our proposed approach is not limited to binary classification, and can be re-purposed by training with a one-class loss that is used for anomaly detection.

**Deep learning** Core challenges for CPD are scaling with more variables and recognizing changes resulting in complex patterns involving many variables. Deep neural networks provide a promising solution for CPD, as they can learn to recognize complex patterns without engineering of features and metrics. CNNs for instance, learn to extract increasingly abstract features from raw data through a stack of non-linear convolutions. This leads to recognition of complex patterns such as hundreds of object types in natural images (Szegedy et al., 2015). RNNs on the other hand, learn complex temporal patterns in sequences of arbitrary length, which is used in applications such as human activity recognition with wearable sensors (Hammerla et al., 2016). These are exactly the type of pattern changes that pose challenges for CPD. On the other hand, a key feature of CNNs is shift-invariance, meaning the prediction will not change even if a pattern shifts in time or space. Gated variants of RNN such as Long Short-Term Memory (LSTM) networks (Hochreiter & Schmidhuber, 1997) and attention-augmented networks (Ba et al., 2015) can also learn shift-invariance, due to their ability to control which part of data to attend or ignore.

Ideally, a CPD method should perform equally well on test data regardless of whether changes happen faster or slower than seen in training data. However, the fixed resolution of CNN and RNN architectures makes them sensitive to scale. CNNs have been extended to model multiple scales simultaneously (Shen et al., 2015), but this is not a scale invariant method, as features are simply concatenated. For RNNs, (Chung et al., 2016) propose a hierarchical architecture to process a sequence through successive RNN layers, at different resolutions. However, layers of RNN there resemble layers of convolution in CNNs (modeling the signal at a different abstraction level) and are not invariant to scale changes at the same abstraction level. Therefore, we propose a new architecture, PRNN, that exploits both CNN and RNN, while augmenting them with scale invariance.

Another limitation of CNNs, and to some extent RNNs for CPD, is the difficulty of modeling long-term dependencies. However, this is necessary to recognize gradual changes. Dilated convolutions have recently allowed long-term dependency modeling in CNNs (Yu & Koltun, 2016; Oord et al., 2016). RNNs are naturally built to model long-term dependencies, but suffer from vanishing gradients. Extensions such as LSTMs and Gated Recurrent Units (GRU) (Cho et al., 2014) solve the problem of vanishing gradients, but still have limited memory space. Intuitively, information from an infinitely long sequence cannot be stored in a fixed-dimensional RNN cell. To reduce the computation complexity for conventional RNN, Campos et al. (2017) proposed Skip RNN to skip state updates while preserving the performance of baseline RNN models. Their skipping-state-updates operation has the advantage of avoiding redundant RNN updates. However, this has the risk of skipping temporal dependencies, especially for long term dependencies, which can hurt the overall performance of RNN. To address this, recent work has augmented RNNs with various types of memory or stack (Sukhbaatar et al., 2015; Joulin & Mikolov, 2015), but these methods are not scale-invariant. Our PRNN, models infinitely long sequences with its multi-scale RNN, which forms a stack of memory cells in an arbitrary number of levels. A higher-level RNN cell in a stack has lower resolution, and thus can store longer dependencies at no additional computational cost, while a lower-level RNN cell has a high resolution and prevents the loss of details in the short term. Frameworks like Feature pyramid networks (Lin et al., 2017) and wavelet CNN (Fujieda et al., 2018) has been proposed to deal with images with different scales or resolutions. However, both of them cannot be applied directly on multivariate time series for change point detection as they cannot model the temporal dependencies in multivariate time series.

## 3 METHOD

We propose a new class of deep learning architectures called Pyramid Recurrent Neural Networks (PRNNs). The model takes a multi-variate time series and transforms it into a pyramid of multi-scale feature maps using a trainable wavelet layer (NWL). All pyramid levels are processed in parallel using multiple streams of CNN with shared weights, yielding a pyramid of more abstract feature maps. Next, we build a multi-scale RNN on top of the pyramid feature map, to encode longer-term, dependencies. The PRNN output is used to detect changes at each time step with a binary classifier.

### 3.1 NEURAL WAVELET LAYER

CNNs can learn to recognize complex patterns in multivariate time series, partly due to parameter-sharing across time (via the convolution operation), which leads to shift-invariance. However, CNNs

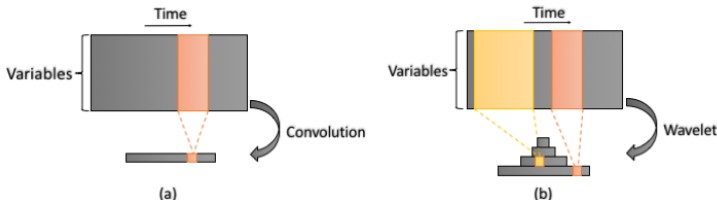

Figure 1: An illustration of (a) a convolutional layer; (b) the Neural Wavelet Layer. Only one feature map of each output has been shown.

are not scale-invariant, so a learned pattern cannot necessarily be recognized when it appears more gradually or more quickly. To augment CNNs with scale invariance, we introduce Deep Wavelet Neural Networks (DWNN), which consist of a proposed Neural Wavelet Layer followed by parallel streams of CNN.

The Neural Wavelet Layer (NWL) can be seen as a set of multi-scale convolutions with trainable kernels, which are applied in parallel on each variable of the input time series. The input to the NWL is a multivariate time series, $X \in \mathbb{R}^{T \times c}$, where $T$ is the number of timepoints and $c$ is the number of variables. The NWL takes $X$ and produces multiple feature maps, which together form a pyramid of convolution responses. That is:

$$f_{NWL}(X) = (H_1, H_2, ..., H_k) : H_i \in \mathbb{R}^{T/2^{i-1} \times c}. \tag{1}$$

An example is shown in Figure 1. Specifically, the NWL uses the filter bank technique **?** for discrete wavelet transform. Given a pair of separating convolutional kernels (typically a low-pass and a high-pass kernel), it convolves the signal with both, outputs the high-pass response, and down-samples the low-pass response for the next iteration. It repeats this process and in each iteration outputs an upper level of the output pyramid. Although traditional wavelets such as Haar or Gabor **?** can be used, we have experimentally found that initializing the filter banks with random numbers and training them using backpropagation with the rest of the network leads to higher accuracy.

More formally, the NWL is characterized by its trainable kernels $K_l^{(v)}, K_h^{(v)} \in \mathbb{R}^{\tau \times c}$ for all variables $v \in \{1...c\}$, where $\tau$ is the kernel size. Given each channel of $X$ as input (e.g. $X^{(v)}$), the NWL iteratively computes lowpass and highpass responses, starting with $L_1^{(v)}$ and $H_1^{(v)}$, that are:

$$L_1^{(v)} = \omega(X^{(v)} * K_l^{(v)}) \quad , \quad H_1^{(v)} = \omega(X^{(v)} * K_h^{(v)}), \tag{2}$$

where $*$ is convolution and $\omega$ is a downsampling operation (e.g. implemented by linear interpolation). At the $i$-th iteration of the wavelet transform, given $L_{i-1}^{(v)}$ and $H_{i-1}^{(v)}$, it computes $L_i^{(v)}$ and $H_i^{(v)}$ such that:

$$L_i^{(v)} = \omega(L_{i-1}^{(v)} * K_l^{(v)}) \quad , \quad U_i^{(v)} = \omega(L_{i-1}^{(v)} * K_h^{(v)}). \tag{3}$$

This operation is repeated for a pre-specified number of times, $k$, or until the length of $L_i^{(v)}$ and $H_i^{(v)}$ becomes smaller than a threshold. The hyperparameter, $k$, can be selected using cross-validation. A larger $k$ (or smaller threshold) results in a larger receptive field at the highest level of the pyramid, enabling the detection of more gradual patterns. However, a large $k$ also brings more computation and also requires a larger buffer in the case of online processing.

The output of each iteration $i \in \{1...k\}$ for variables $v \in \{1...c\}$ can be concatenated to form

$$L_i = [L_i^{(1)}|L_i^{(2)}|...|L_i^{(c)}] \quad , \quad H_i = [H_i^{(1)}|H_i^{(2)}|...|H_i^{(c)}], \tag{4}$$

where $[.|.]$ indicates concatenation. The output of the NWL is the stack of all $H_i$. These are called different *levels* of a *pyramid* throughout this paper. In the original filter bank method the last lowpass response, $L_k$, is also stacked with the output but we did not observe an improvement with $L_k$.

The key advantage of a NWL over a conventional convolution layer is that a single wavelet can encode the input with multiple granularities at once, whereas a single convolution only encodes a single granularity. Although different layers of a CNN have different granularities, they encode the

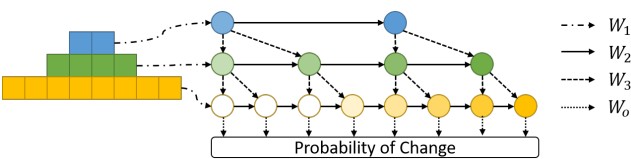

Figure 2: An illustration of the proposed Pyramid Recurrent Layer, with downsampling ratio of 2.

data at a different level of abstraction, and thus cannot simultaneously extract the same pattern at different scales. On the other hand, a single wavelet layer can encode changes with the same patterns at different paces, simultaneously into the same feature map, at different levels of the pyramid.

We will use the proposed NWL as a part of a larger, deeper architecture, which is described in the rest of this section. Hence, an important aspect of NWL is that it can be used as a layer of a deep network, in composition with other neural layer types such as convolutional and fully connected layers. For example, the input to a wavelet layer can be the output of a convolutional layer. Alternatively, to stack a convolutional layer on the output of a wavelet layer, one should apply the convolution on each level of the wavelet pyramid, resulting in a pyramid-shaped output.

Accordingly, a network composed of one wavelet layer and an arbitrary number of other layers, can take a multi-variate time series as input, and produce a pyramid-shaped response as output. We refer to such a network architecture as a Deep Wavelet Neural Network (DWNN). In this paper we use a specific form of DWNN, which starts with a NWL, directly applied on the input time series $X$, followed by parallel streams of CNN with shared parameters, each of which takes one level of the NWL pyramid. More specifically, we use an $\ell$-layer CNN with a down-sampling stride of $p_j$ at the $j$-th layer, which results in a total down-sampling factor of $P = \prod_{j=1}^{\ell} p_j$, and with $f_j$ feature maps at the $j$-th layer. We apply that CNN in parallel on each level of the output pyramid of the NWL, which means for each $i \in \{1...k\}$, it gets $H_i \in \mathbb{R}^{T/2^{i-1} \times c}$ and outputs $C_i \in \mathbb{R}^{T/2^{i-1}/P \times f_\ell}$.

### 3.2 Pyramid Recurrent Layer

The output of the DWNN is a multi-scale pyramid of sequential feature maps that encode short-term temporal patterns at different times and scales. It is common to process sequential features using an RNN, to encode longer-term temporal patterns. However, conventional RNNs process a single sequence, not a multi-scale pyramid of sequences. Similar to the need for a wavelet layer, RNNs are not scale-invariant, meaning if an RNN can recognize a pattern, it does not necessarily imply it can recognize a temporally shortened or stretched instance of the same pattern without having seen this scale in the training data. Further, RNNs fail to learn very gradual patterns, due to limited memory. While this can be addressed by memory-augmented networks, they remain sensitive to scale.

To address these issues, we introduce a novel hierarchically connected variant of RNNs. Our proposed network, PRNN, scans the multi-scale output of a DWNN, and simultaneously encodes temporal patterns at different scales. An RNN is applied in parallel on different levels of the input pyramid. On each level at each step, it takes as input the corresponding entry from the input pyramid, along with the most recent output of the RNN operating at the upper level. We concatenate those two vectors and feed as input to the RNN. We refer to this technique as Pyramid Recurrent Layer (PRL).

Denoting the value at level $i$ of the input pyramid at time $t$ as $C_i[t]$, and assuming the downsampling ratio in wavelet transform is $d$, (i.e., each level of the pyramid has $d$-times the length of its upper level) we can write the recurrent state at level $i$ and time $t$ as:

$$h_i[t] = \sigma(W_1 C_i[t] + W_2 h_i[t-1] + W_3 h_{i+1}[\lfloor t/d \rfloor] + b), \tag{5}$$

where $\sigma$ is a nonlinear activation function such as ReLU, and $W_1$, $W_2$, $W_3$ and $b$ are trainable parameters of this layer. These parameters define a linear transformation of the current state, past state, and higher-level state, as illustrated in Figure 2. Note that the proposed hierarchical structure is agnostic of the function of each cell. Although we used a simple RNN cell for illustration, we could use any variant of RNNs such as a Long Short-Term Memory (LSTM) (Hochreiter & Schmidhuber, 1997) or Skip RNN (Campos et al., 2017) as our RNN cell.

The proposed architecture can be compared with an RNN operating on a single data sequence. If the data granularity is high, the RNN likely fails to model long-term dependencies, due to the well-known problem of vanishing gradients. One can lower the data granularity, so long-term patterns can be summarized in fewer steps, but this results in the loss of details. Accordingly, conventional RNNs were not designed to effectively detect both abrupt and gradual patterns at the same time.

On the other hand, in the proposed PRL, each RNN unit is provided with inputs from the same level of granularity as well as the level above. The RNN that operates at the lowest level, in turn, receives information from all levels of granularity. Figure 2 illustrates the effect of forgetting using decreasing color saturation. While it is impossible to keep track of the past through the lower level alone, the information path from upper levels connect the past to present in only three steps. This lets the PRL model long-term patterns, while it can still model details through the lower levels.

### 3.3 Pyramid Recurrent Neural Networks

We propose PRNN as a composition of a DWNN and a PRL. An input time series of arbitrary length is transformed through a DWNN into a pyramid-shaped representation, which is then fed into a PRL. For CPD and other classification problems, a logistic regression layer is built on the output of the RNN cells that operate at the lowest level of the pyramid. This layer produces detection scores at each time step with the highest possible granularity. Specifically, the detection score for time $t$ is:

$$y_t = \sigma(W_o h_1[t] + b_o), \tag{6}$$

where $\sigma$ is the sigmoid function and $W_o$ and $b_o$ are trainable parameters. The classification loss at each time is the cross entropy loss written as:

$$E_t = y_t^* \log y_t + (1 - y_t^*) \log (1 - y_t), \tag{7}$$

where $y_t^*$ is the ground truth at time $t$. We optimize this loss using stochastic gradient descent on parameters of the classifier ($W_o$ and $b_o$), PRL ($W_1$, $W_2$, $W_3$ and $b$), and NWL ($K_l$ and $K_h$).

## 4 Evaluation

We compare the proposed PRNN to conventional deep learning baselines. Using both simulated and real-world datasets, we show that PRNNs can detect abrupt and gradual changes more accurately than baseline approaches and can be used for activity recognition by learning labels for different changes.

### 4.1 Datasets

**Synthetic dataset** We create a synthetic dataset to evaluate accuracy at simultaneously detecting gradual and abrupt changes. We construct 2000 time series each with 12 variables and 8192 time steps (a power of two chosen to avoid rounding errors in downsampling). Each time series is a combination of a Brownian process and white noise and has 4 changepoints at randomly chosen times. A change is a shift in the mean of 4 randomly chosen dimensions, with randomly chosen speed (duration of change) and amount of shift. A speed of 0 gives an abrupt change, while longer ones provide more challenging cases to recognize. An example of the simulated time series together with ground truth and detection results are shown in Figure 3. We randomly split the data in half, 1000 for training and 1000 for testing. To demonstrate robustness of the proposed method against variability in scale, we also do a split by scale, where all changes in one half are strictly more gradual than all in the other half.

**Opportunity dataset** For real-world evaluation, we first use the OPPORTUNITY activity recognition dataset (Chavarriaga et al., 2013), which consists of on-body sensor recordings from 4 participants performing activities of daily living, such as cleaning a table. Each participant has 6 records (runs) of around 20min each. Values of 72 sensors from 10 modalities were recorded at 30Hz, and manually labeled with 18 activity types. Following (Hammerla et al., 2016), we ignore variables with missing values, which leads to 79 variables for each record. We use run 2 of subject 1 for validation and runs 4 and 5 of subjects 2 and 3 for test, and the rest for training. To repurpose this activity recognition dataset for CPD, we consider the transition between two activities as a change. This transition can take place at various durations, which makes the task challenging. As ground

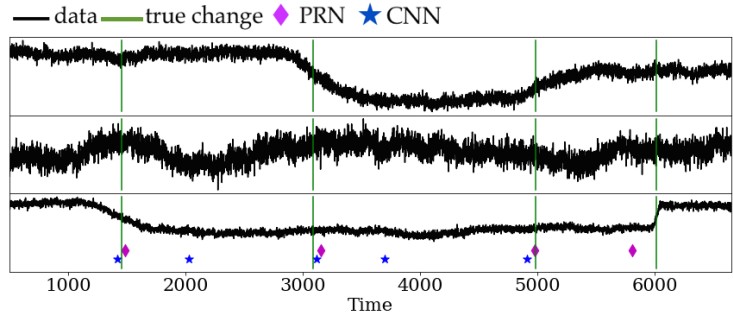

Figure 3: Detected changes in 3 of 12 dimensions of the synthetic dataset.

truth, we use the temporal annotation provided with the OPPORTUNITY dataset to determine moments that the activity type changes. Fig. 6 shows a sample of this dataset with ground truth and detection results.

**Bee Waggle Dance dataset** We also test our methods on the Bee Waggle Dance data (Oh et al., 2008). Honey bees perform waggle dance to communicate with other bees about the orientation and distance to the food sources. The Bee Waggle Dance data includes six videos of bee waggle dances with 30 frames per second. The data include 3 variables encoding the honey bee's position and head angle at each frame. Using the position and angle information, each frame is labeled with activity of "turn left", "turn right", or "waggle dance." Similar to the OPPORTUNITY dataset, we consider the transition between two activities of the honey bee as a change point. We test our method and other baselines on "sequence 1" of the bee data. We train on the first 256 frames (a power of 2 chosen to avoid rounding errors) and test on the other 768 frames. We use small size of training data to see how the proposed method behaves and for consistency with other prior works (Saatçi et al., 2010).

## 4.2 BASELINES

We compare the proposed architecture to the following unsupervised CPD method and supervised deep-learning baselines:

**GGM** Xuan & Murphy (2007) is related to BOCPD (Adams & MacKay, 2007), a classic method for CPD, but was selected to provide fairer comparison against our approach as it is offline and incorporates multivariate time series.

**CNN** We use a CNN that takes a time series as input and predicts a sequence of detection scores for changes. Due to the widely used max-pooling layers, the output has a lower temporal granularity compared to the input. We denote the ratio of output length to the input length as $\gamma$.

**RCN** We apply an RNN to the output of the CNN. The output has the same granularity as CNN, while each step of the output has a larger receptive field that encodes all the past data.

**DWNN** We use the proposed DWNN, which is formed by applying an NWL to the input time series and feeding the output pyramid levels to parallel branches of a CNN. The output of CNN branches are upsampled to have the same size and fused by arithmetic mean.

**PRNN** We apply the complete proposed method which consists of a DWNN followed by a Pyramid Recurrent Layer to fuse levels of the pyramid.

**PRNN-S** As a final baseline, we replace the conventional RNN cell in our PRNN method with a Skip RNN (Campos et al., 2017) which was found to have lower time complexity. This enables us to test whether an efficient RNN can preserve the performance of PRNN.

## 4.3 IMPLEMENTATION DETAILS

All of the deep-learning baselines share a core CNN architecture on which the additional modules are built. We fix the architecture of the core CNN to be $[9:128:4], [5:128:2], [5:128:2]$, where we use the notation $x:y:z$ for a convolution layer where $x$ is the kernel size, $y$ is the number of output

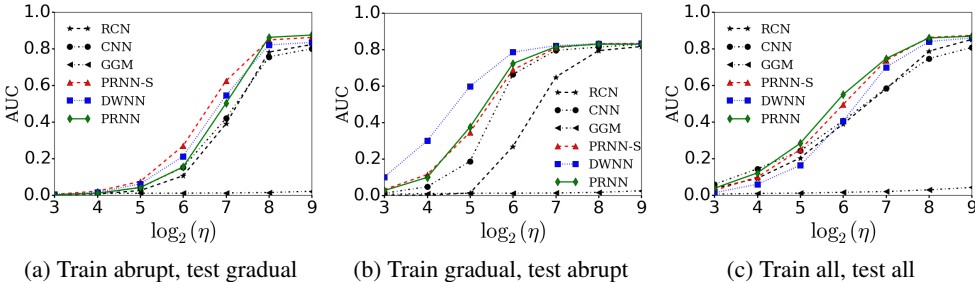

(a) Train abrupt, test gradual     (b) Train gradual, test abrupt     (c) Train all, test all

Figure 4: AUC (Area Under the ROC Curve) results for the synthetic dataset with three different test/train scenarios. $\eta$ is the tolerance for how close in time a detected change must be to a true change to be considered a positive. See Appendix A for AUC score tables.

feature maps, and $z$ is the pooling stride. Each convolution layer is followed by max-pooling and ReLU activation. The output of all baselines are fed to a fully connected perceptron with sigmoid activation which results in binary detection scores at each time step. The granularity ratio $\gamma$ for this architecture is $1/16$. For DWNN, PRNN, and PRNN-S, we used a 7-level wavelet with kernel size 3 for both synthetic and OPPORTUNITY dataset. For Bee Waggle Dance data, due to the small size of the data and the more abrupt activities (compared with synthetic and OPPORTUNITY dataset) of honey bee, we used a 5-level wavelet with kernel size 3. For all datasets RCN and PRNN used an LSTM cell with 256 hidden units.

We train all models using Adam (Kingma & Ba, 2014) with early stopping to avoid overfitting. At test time, the models take a time series and predict a sequence of detection scores. To detect changepoints, we apply non-maximum suppression with a sliding window of length $\omega$ and filter the maximum values with a threshold. We evaluate AUC by iterating over this threshold. Hyper-parameter $\omega$ controls how nearby two distinct changes can be detected and is tuned for each method separately using cross-validation.

The real world datasets (Bee data and OPPORTUNITY data) are more challenging than the synthetic data, as they include diverse changepoints formed by transitions between many activity types. To address this, we use multitask learning, training the model to both detect changes and classify activity by changing the output dimension of the last fully connected later to have multiple units (19 for OPPORTUNITY data, and 4 for Bee data). For OPPORTUNITY data, the first 18 units predict a log probability for each activity and the last 1 unit outputs the probability of a change point (for bee data, it's 3 units and 1 unit). We define a softmax cross-entropy loss on those 18 units and add it as a regularization term to the objective function. Multitask learning improved the results equally for all baselines, because the model has auxiliary information, namely the activity type and not just the existence of a change.

For GGM, we use the full covariance model instead of the independent features model to capture the correlations between features. We use a uniform prior as in (Xuan & Murphy, 2007), and set the pruning threshold to $10^{-20}$. Since there is no training for GGM, we evaluate the algorithm using the same test data as all other methods we compared on both synthetic and real world dataset.

We evaluate precision and recall, and report AUC. As detected changepoints may not exactly match the true changepoints, we use a tolerance parameter $\eta$ that sets how close a detected change must be to a true change to be considered a correct detection. We match detected changepoints to the closest true changepoint within $\eta$ time steps. Precision is the number of matched detections divided by the number of all detections, and recall is the number of matches divided by the number of true changes.

## 4.4 RESULTS

### 4.4.1 SYNTHETIC DATA

For the synthetic dataset, three different train/test splits were used to demonstrate extrapolation from gradual to abrupt changes and vice versa. Fig. 4c shows the results for a random split (mixing

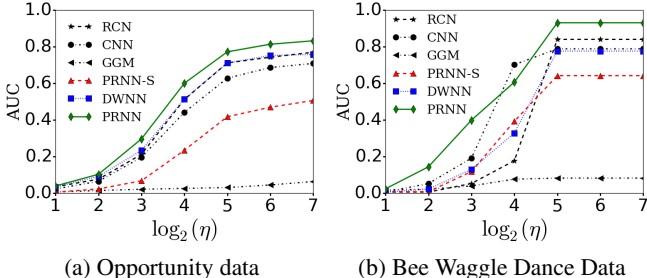

(a) Opportunity data        (b) Bee Waggle Dance Data

Figure 5: AUC (Area Under the ROC Curve) results for Opportunity and Bee Waggle Dance data. $\eta$ is the tolerance with a unit of 1/30 seconds for both dataset. See Appendix B for AUC score tables.

scales), while Fig. 4a and 4b use the scale-variant split introduced in section 4.1. In the scale-variant split, the model needs to extrapolate patterns learned from training data to scales that have not been observed. This is extremely challenging for a model that is not scale-invariant. This is apparent in Fig. 4a where both CNN and RCN show worse results in parts compared to their own performance in Fig. 4c. From experiment of Fig. 4c to Fig. 4a, AUC of CNN (RCN) decreased from 41% (39%) to 15% (11%) when the tolerance is 64 steps ($2^6$). This is because the methods are not designed to be specifically robust against scale variability. In the transition from 4c to 4a, DWNN, PRNN, and PRNN-S like other methods inevitably suffer from a performance drop, which is due to the increase in task difficulty. However, the amount of this drop is substantially lower for DWNN, due to the wavelet layer and shared parameters across scales. At a tolerance of 64 steps, for instance, the performance drop for DWNN is 20%, which is lower than PRNN-S (22%) and CNN (26%). Again DWNN and CNN respectively work better than PRNN and RCN in this setting, which is consistent with the overall results (See Appendix A for AUC details).

While recognizing abrupt changes from gradual training ones (Fig. 4b) is easier than recognizing a mix of scales, CNN and RCN perform worse than our approach due to their inability to generalize in scale. In Fig. 4b, when tolerance is 64 steps, AUC for CNN and RCN are 66% and 30%, which are lower than both PRNN (72%) and DWNN (79%). In contrast, DWNN, PRNN-S, and PRNN have higher AUC than their own performance in the mixed experiment (Fig. 4c). The performance of PRNN at different tolerances is 20-25% higher on average in the mixed experiments than the experiment of "train abrupt, test gradual" (Fig. 4a). In the train on gradual and test on abrupt experiment (4b), DWNN performs even better than PRNN and PRNN-S in all tolerances, and similarly, CNN outperforms RCN. This shows recurrent architectures are generally less effective for this kind of extreme generalization. The high performance of DWNN 4b also shows the effectiveness of the added wavelet layer in modeling both gradual and abrupt changes in time series. However, in real-world cases we are more likely to have a mix of scales in both training and test, and it is in this case (fig. 4c) that PRNN is most accurate. As shown in the AUC plots, it is in general more difficult to recognize gradual changes. It is possible to adapt our work to detect segments rather than specific points (e.g. as in (Bardwell & Fearnhead, 2017)), if instead of applying a non-maximum suppression on the output score map of change, we perform binary segmentation to detect intervals with continuously high detection score.

Figure 3 shows example results for our scale invariant PRNN and scale sensitive CNN. Overall CNN has a higher false positive rate, while also missing one of the changes. While detected changes and ground truth are not always precisely aligned, the small gaps are acceptable in the case of gradual changes, where it can be hard to define a single moment when the change occurs.

### 4.4.2 OPPORTUNITY DATA

Figure 6 shows results and AUC plots for the OPPORTUNITY dataset. In the time series, we see that CNN has a missed detection and at least one false positive around time 300, while PRNN detects all changes close to their actual times. In fig 5a, we see that PRNN outperforms other methods at all tolerance levels. In contrast to the synthetic data, PRNN-S has significantly lower AUC than both PRNN and DWNN for every tolerance. It may be that Skip RNN is skipping important information

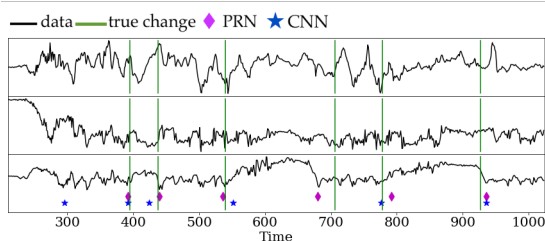

Figure 6: Detected changes on sample (3 of 79 dimensions shown) of Opportunity dataset.

encoded in our wavelet later. Finally, the performance of GGM is lowest for all cases. This is not surprising, as it is an unsupervised method, and does not learn from previously observed patterns. When the tolerance is 64 (around 2 seconds, $\eta = 2^6 = 64$), a reasonable value for practical activity recognition use, PRNN achieves 81% AUC while DWNN, RCN, CNN, and PRNN-S respectively achieve 75%, 74%, 69%, 47%. Full results can be seen in Appendix B.1.

The five deep learning methods, PRNN, PRNN-S, RCN, DWNN, and CNN, respectively took 110, 105, 80, 24, and 6 minutes to train and converge on the OPPORTUNITY dataset. Recurrent methods generally take longer due to backpropagation through time. However, this only happens during training, and does not affect test complexity. One can compare PRNN to RCN, and DWNN to CNN, and observe an increase in time complexity. This is due to repeating computations on multiple levels of a pyramid. This however, only multiplies the time complexity by a constant factor, since the length of pyramid levels exponentially vanish. Note that DWNN has a superior performance to RCN in most cases, while also being faster to train.

### 4.4.3 BEE WAGGLE DANCE DATA

Figure 5b shows AUC plots for all methods we tested on Bee Waggle Dance dataset. Our PRNN method outperforms other methods when the value of $\eta$ is no less than 5 (around 1 second) with AUC of 93%. Similar to the result on OPPORTUNITY dataset, GGM has the lowest AUC for all tolerances. When the tolerance is 64 (around 2 seconds, $\eta = 2^6 = 64$), PRNN achieves 93% AUC while PRNN-S, RCN, and CNN respectively achieve 64%, 84%, 78% (see Appendix B.2 for AUC details). Similar to OPPORTUNITY dataset, the drop of AUC for PRNN-S is caused by the skipping of states updates. However, compared with the OPPORTUNITY data where PRNN-S has maximum AUC of 51%, PRNN-S for Bee Waggle Dance data has higher maximum AUC of 64%. This is because the changes in honey bee activities are more abrupt than human activities, so the skipped updates have lower impact on the detection performance. From the AUC plots, a change in tolerance affects our PRNN much less compared to other methods. For instance, when the tolerance is lowered from 32 ($\eta = 2^5 = 32$) to 16 ($\eta = 2^4 = 16$), the AUC of RCN drops significantly (from 84% to 18%), while AUC of PRNN drops much less (from 93% to 61%). CNN has a dramatic drop in accuracy from $\eta = 4$ to $\eta = 3$, suggesting it is consistently detecting changes with a delay. Thus, PRNN is less sensitive to this parameter and more reliable for real world cases.

## 5 CONCLUSION

We propose a new class of DNNs that are scale-invariant, and show they can detect from abrupt to gradual changepoints in multimodality time series. The core is 1) augmenting CNNs with trainable Wavelet layers to recognize short-term multi-scale patterns; and 2) building a pyramid-shaped RNN on top of the multi-scale feature maps to simultaneously model long-term patterns and fuse multi-scale information. The final model can detect events involving short- and long-term patterns at various scales, which is a difficult task for conventional DNNs. Although this reduces the amount of training data required to learn from changes, the proposed method still requires clean labels. Experiments show our approach detects changes quickly, with lower sensitivity to the tolerance parameter than other approaches. For real-world applications, this leads to much higher reliability. In future work we will real-world challenges (e.g. noisy data, missing/noisy labels) by incorporating robustness, semi-supervised learning methods, and multi-view learning techniques.

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

## APPENDIX A   SYNTHETIC DATA

Table 1-3 show the AUC (Area Under the ROC Curve) results for synthetic data. To detect change-points, we apply non-maximum suppression with a sliding window of length $\omega$ and filter the maximum values with a threshold. We evaluate AUC by iterating over this threshold. Since changepoints may not exactly match the true changepoints, we use a tolerance parameter $\eta$ that sets how close a detected change must be to a true change to be considered a correct detection. We match detected changepoints to the closest true changepoint within $\eta$ time step.

Table 1 shows the results for the experiment of "train abrupt and test gradual" for synthetic data. Table 2 shows the results for the experiment of "train gradual and test abrupt" for synthetic data. Table 3 shows the results for the experiment of "train all and test all" for synthetic data.

Table 1: AUC results for the experiment of "train abrupt and test gradual" for synthetic data for each tolerance ($\eta$)

| Tolerance ($\eta$) | RCN | CNN | GGM | PRNN-S | DWNN | PRNN |
|---|---|---|---|---|---|---|
| 8 | 0.002 | 0.003 | 0.007 | 0.006 | 0.005 | 0.003 |
| 16 | 0.008 | 0.011 | 0.010 | 0.023 | 0.016 | 0.011 |
| 32 | 0.026 | 0.042 | 0.012 | 0.076 | 0.063 | 0.043 |
| 64 | 0.107 | 0.153 | 0.012 | 0.271 | 0.213 | 0.155 |
| 128 | 0.391 | 0.421 | 0.013 | 0.625 | 0.546 | 0.503 |
| 256 | 0.783 | 0.757 | 0.016 | 0.849 | 0.822 | 0.863 |
| 512 | 0.824 | 0.801 | 0.023 | 0.862 | 0.835 | 0.876 |

Table 2: AUC results for the experiment of "train gradual and test abrupt" for synthetic data for each tolerance ($\eta$)

| Tolerance ($\eta$) | RCN | CNN | GGM | PRNN-S | DWNN | PRNN |
|---|---|---|---|---|---|---|
| 8 | 0.001 | 0.013 | 0.007 | 0.033 | 0.102 | 0.027 |
| 16 | 0.003 | 0.049 | 0.011 | 0.115 | 0.301 | 0.100 |
| 32 | 0.014 | 0.188 | 0.013 | 0.347 | 0.599 | 0.376 |
| 64 | 0.269 | 0.665 | 0.013 | 0.689 | 0.787 | 0.724 |
| 128 | 0.650 | 0.797 | 0.014 | 0.811 | 0.822 | 0.816 |
| 256 | 0.795 | 0.814 | 0.018 | 0.833 | 0.833 | 0.830 |
| 512 | 0.817 | 0.830 | 0.026 | 0.834 | 0.834 | 0.830 |

Table 3: AUC results for the experiment of "train all and test all" for synthetic data for each tolerance ($\eta$)

| Tolerance ($\eta$) | RCN | CNN | GGM | PRNN-S | DWNN | PRNN |
|---|---|---|---|---|---|---|
| 8 | 0.039 | 0.061 | 0.007 | 0.027 | 0.014 | 0.039 |
| 16 | 0.093 | 0.144 | 0.011 | 0.100 | 0.061 | 0.122 |
| 32 | 0.204 | 0.244 | 0.012 | 0.249 | 0.164 | 0.284 |
| 64 | 0.390 | 0.407 | 0.017 | 0.496 | 0.406 | 0.551 |
| 128 | 0.582 | 0.586 | 0.021 | 0.737 | 0.700 | 0.747 |
| 256 | 0.788 | 0.747 | 0.030 | 0.863 | 0.840 | 0.860 |
| 512 | 0.852 | 0.808 | 0.043 | 0.874 | 0.860 | 0.869 |

## APPENDIX B    REAL WORLD DATA

### B.1    OPPORTUNITY DATA

Table 4 shows the results for Opportunity data.

Table 4: AUC results of Opportunity data for each tolerance ($\eta$)

| Tolerance ($\eta$) | RCN | CNN | GGM | PRNN-S | DWNN | PRNN |
|---|---|---|---|---|---|---|
| 2 | 0.036 | 0.024 | 0.007 | 0.007 | 0.034 | 0.040 |
| 4 | 0.077 | 0.066 | 0.016 | 0.024 | 0.093 | 0.104 |
| 8 | 0.213 | 0.197 | 0.022 | 0.068 | 0.234 | 0.297 |
| 16 | 0.513 | 0.442 | 0.027 | 0.236 | 0.515 | 0.601 |
| 32 | 0.713 | 0.629 | 0.032 | 0.418 | 0.712 | 0.773 |
| 64 | 0.744 | 0.687 | 0.046 | 0.471 | 0.753 | 0.815 |
| 128 | 0.771 | 0.710 | 0.065 | 0.507 | 0.759 | 0.833 |

### B.2    BEE WAGGLE DANCE DATA

Table 5 shows the results for Bee Waggle Dance data.

Table 5: AUC results of Bee Waggle Dance data for each tolerance ($\eta$)

| Tolerance ($\eta$) | RCN | CNN | GGM | PRNN-S | DWNN | PRNN |
|---|---|---|---|---|---|---|
| 2 | 0.007 | 0.008 | 0.019 | 0.009 | 0.007 | 0.025 |
| 4 | 0.007 | 0.053 | 0.023 | 0.009 | 0.023 | 0.145 |
| 8 | 0.054 | 0.192 | 0.041 | 0.119 | 0.131 | 0.400 |
| 16 | 0.178 | 0.703 | 0.077 | 0.393 | 0.329 | 0.608 |
| 32 | 0.841 | 0.789 | 0.083 | 0.643 | 0.777 | 0.932 |
| 64 | 0.841 | 0.789 | 0.083 | 0.643 | 0.777 | 0.932 |
| 128 | 0.841 | 0.789 | 0.083 | 0.643 | 0.777 | 0.932 |

