# OpenReview forum: "Pyramid Recurrent Neural Networks for Multi-Scale Change-Point Detection"
_ICLR.cc/2019/Conference_

### Official Review · AnonReviewer2 · 2018-11-01
**Interesting deep architecture for multi-variate time series modeling, lacks proper comparison with existing literature**

**Rating:** 7
**Confidence:** 4

**Review:**

1. This papers leverages the concept of wavelet transform within a deep architecture to solve the classic problem (especially for wavelet analysis) of change point detection. The authors do a reasonably comprehensive job of demonstrating the efficacy of the proposed framework using various synthetic and real data sets with both gradual and abrupt changes

2. The concept of pyramid network idea is not really new, in the context of CNN it has been established quite well. The paper should highlight this fact by citing papers such as "Lin, Tsung-Yi, et al. "Feature Pyramid Networks for Object Detection." CVPR. Vol. 1. No. 2. 2017."

3. Involving wavelet transforms in deep nets have been done before. This paper attempts to learn wavelet transform parameters by involving them as trainable layers. But even this kind of idea is also emerging in the community. Papers such as "Fujieda, Shin, Kohei Takayama, and Toshiya Hachisuka. "Wavelet Convolutional Neural Networks." arXiv preprint arXiv:1805.08620 (2018)" need to be discussed in this context.

4. The biggest issue in my mind is that I feel "Chung et al 2016" is still a very similar framework as the proposed one. While authors argue that it uses more like CNN architecture and the proposed method may pick up the multi-scale features better, comparison with this seems to be most appropriate. This will also clearly identify the benefits of the wavelet structure to the filters and multi-resolution analysis approaches.

5. RCNN term has been used for CNN+RNN architecture. This may not be a good terminology to use since RCNN is a very popular term referring to Region based CNN for detection and localization purposes.

6. AUC metric, I believe is the - area under ROC curve, this needs to be spelled out, how it is computed? at least in the Appendix

xxxxxxxxxxxxxxxxxxx

Appreciate the authors' rebuttal, updated my score.

---

> ### Author Response · Authors · 2018-11-16
> **Comparison with existing literature and implementation details**
>
> We thank the reviewer for their helpful comments and have both updated the paper and planned experiments to further improve it. Specific replies are below.
>
> 2-3) The reviewer states that “the concept of pyramid network is not really new” and that “involving wavelet transforms in deep nets have been done before”. We have added both papers to the related work, however there are crucial differences between our approach and the cited works. Both Lin et al and Fujieda et al address multiscale data, though this is in the context of image data with varying scale/resolution. However, these works focus on images, each of which has data at multiple scales. In contrast, we focus on temporal dependencies among multivariate time series, which are not captured by these approaches. For example, Lin et al. built feature pyramids for each input image to model different scales, but in our data the change points relate to earlier signals rather than the same moment captured at a different resolution. For example, blood glucose changes in response to factors like exercise or food intake that happened earlier. The wavelet method proposed by Fujieda et al has the same limitations, since that method is designed specifically for capturing spatial dependencies for different scales.
>
>
> 4) The reviewer had some questions about how the paper advances the state of the art over the work of Chung et al (2016). While that work also examines temporal dependencies at different scales, the approach is not designed specifically for CPD. However, this approach is not scale invariant, relies on the hierarchical boundary structure, and has stronger connections between layers in the hierarchy. The available code was used for prediction, but we are now attempting to apply the approach to the datasets in our paper by replacing the RNN framework in our proposed PRNN with the Hierarchical Multiscale Recurrent Neural Network (HMRNN) proposed by Chung et al. The comparison results will be added to our paper once the experiments are completed.
>
>
> 5) We have updated the terminology used.
> 6)AUC: We included detail on the calculation under the implementation details (sec. 4.3) of our original submission, but have revised to make this clearer and have expanded the appendix to discuss how AUC is computed for each tolerance level.
>
> To detect changepoints, we apply non-maximum suppression with a sliding window of length $\omega$ and filter the maximum values with a threshold. We evaluate AUC by iterating over this threshold. Since changepoints may not exactly match the true changepoints, we use a tolerance parameter $\eta$ that sets how close a detected change must be to a true change to be considered a correct detection. We match detected changepoints to the closest true changepoint within $\eta$ time step

---

### Official Review · AnonReviewer1 · 2018-11-01
**Nice model with somewhat lacking experimental validation.**

**Rating:** 6
**Confidence:** 3

**Review:**

This paper proposes a pyramid based neural net which both decomposes a signal into several scales (learning the basis functions to do that) and processes the resulting bands in a scale invariant manner. The method is applied to 1D signals with underlying processes occurring at different time scales where the task is change point detection.

Pros:
* Nice model and model formulation - learning the basis functions both for the low and high frequency is a nice idea. I also liked the way weights are shared across scales. In particular that the information flow between consecutive scales is shared, as well as through time.

* Writing is very clear and method is well motivated

Cons:
* I found the experimental validation a bit limited - the presented results are nice and for the problem quite comprehensive but I would have wanted something a bit more complicated than change point detection. Specifically, since the natural world is full of scale free phenomena it would have been much more interesting with other tasks (generative models? natural images? many options). I feel this would have made the case for the paper much stronger.
* There's also very little analysis of what is learned from the data - how do the kernels look like and how do they correspond to known wavelets? to the data? would be nice to understand what's going on here.


Bottom line:
I like the proposed model and for what it is it's quite good but it would have been a much more convincing paper with more experiments demonstrating the power of the method and analyzing it.

---

> ### Author Response · Authors · 2018-11-15
> **Experimental validation and insight**
>
> We thank the reviewer for their comments on the approach. The review raised two main concerns: scope of experimental validation, and insight into the results.
>
> Experimental validation: while we believe the models are highly general, we note that changepoint detection (CPD) is a fairly large area of ongoing work, without a dominant method that is widely applicable in realistic cases. Thus while we agree that potentially we could have evaluated on other classification problems, we believe that advances in CPD are significant. One point of evidence for this is that the original BOCPD paper by Adams & McKay has 341 citations. However, in future work we plan to test our approach on other data types such as images.
>
> Results: The reviewer stated  “there's also very little analysis of what is learned from the data - how do the kernels look like and how do they correspond to known wavelets? to the data? would be nice to understand what's going on here”
>
>
> The number of wavelets affects how much temporal dependencies are modeled, which then affects the performance of our method. We find that when the change points are more abrupt we need fewer wavelets because the temporal dependencies are shorter for abrupt changes. Therefore, we used a 7-level wavelet with kernel size 3 for both synthetic and OPPORTUNITY dataset but used a 5-level wavelet with kernel size of 3 for Bee Waggle dance data (discussed in implementation details in section 4.3). This is because the changes in honey bee activities are more abrupt than human activities. Another reason for lowering the number of wavelets is due to the small amount of training data. We use fewer wavelets when the training data is smaller to avoid overfitting. Thus, we use 7-level wavelets for synthetic and OPPORTUNITY data but 5-level wavelet for Bee Waggle Dance data.

---

### Official Review · AnonReviewer3 · 2018-11-06
**Could be interesting but biased in many aspects**

**Rating:** 4
**Confidence:** 5

**Review:**

The paper presents an interesting approach to change point detection. I agree we need more general model to capture the change. However, unfortunately, they did not place the contribution correctly with respect to existing literature. The comments for prior work seem to be highly biased. For instance,  in Section 2, "these methods either have unrealistic assumptions, such as defining changes as a large difference in covariance matrix". I would like to comment that, covariance change can capture a large number of changes in real applications and these are not unrealistic assumptions.

The "pyramid" recurrent neural network seems to be a extension of RNN using the idea of multi-scale structure. Could be interesting.

The paper gives too much emphasis on the "merit" of the neural networks on capturing the change patterns. However, there is a very important aspect been ignored or hiding: in order to train neural networks to capture anomaly patterns, since neural networks are highly over-parameterized model, usually there won't be a large number of samples for anomalies. Therefore, in many situations, it is simply unpractical to train neural networks to capture post-change samples.

There is a large body of literature on change point detection in statistics etc. (the author mentioned one, Chen and Zhang 2015, more over, the comment that "they can only detect abrupt change" is wrong, the method is quite general).

The paper fails to have any comparison with existing methods. For instance, how does the proposed method compare with hoteling T-square statistic, or CUSUM statistic, or generalize likelihood ratio statistic, or MMD statistic (non-parametric approach)? Without any comparison, it does not make sense to claim proposed method is superior.

---

> ### Author Response · Authors · 2018-11-15
> **Relation of proposed approach to existing literature**
>
> We appreciate the reviewer’s comments and the opportunity to clarify some areas of question on related work and the amount of training data needed.
>
> Relation of proposed approach to existing literature: We have updated the related work and respond to specific questions and concerns here as well. The reviewer noted that “covariance change can capture a large number of changes in real applications” and felt the assumptions were realistic. We agree that these methods can capture a wide array of changes, but they still require changepoints to result in changes in the covariance matrix. However, in the applications we focus on, this is not always true. Two key examples are when a change involves multiple dimensions and when changes are gradual. First, in activity recognition, multiple sensors with multiple dimensions (x,y,z axis) are often used. A change in activity may only result in a small difference on each sensor individually, while the change is easier to detect by examining differences across the set of sensors. Second, we focus on detecting both abrupt and gradual changes. A gradual change, such as weight loss, will be difficult to detect using the covariance matrix since the change in value will not be large at each individual time.
>
> Chen & Zheng (2015) developed a highly general graph-based method that can be applied to data with arbitrary dimension. However, this method relies on the assumption that the data will have different probability distributions after change points occur. The approach quantifies the similarities among different distributions using a graph with edges showing how “close” they are. Yet in real world settings, when a change point occurs over a long duration of time (e.g. the develop of a chronic disease takes years), the variation of the underlying data distribution will be so small in a short time range that it is undetectable.
>
> More generally, the reviewer asked about how the approach compares to the wide range of statistical methods for changepoint detection (e.g. hotelling T-square, CUMSUM, etc). These methods detect change points when the chosen statistics indicate significant variations for the data. However, their performance is heavily dependent on the chosen parameters the data distribution meeting the assumptions of each method. For example, MMD relies heavily on the choice of kernels. Both and hotelling-square and MMD are significantly affected by the choice of parameters. Further, Hotelling T-square is optimal for multivariate Gaussian case. but not all data is multivariate Gaussian. In real world cases, data can be non-Gaussian for reasons like skewed by outliers or following a different distribution (e.g. exponential). For example, medical is often skewed due to biased samples (e.g. no data from healthy populations when studying ICU patients), and may contain significant outliers due to noise and measurement/device error. Another example is that when the data (e.g. blood glucose) is affected by multiple factors (e.g. exercise, eating) that follow different distributions, the data will also not be Gaussian. Conventional CUSUM is optimal in detecting small consistent changes and relies heavily on the prior information of the changing parameters that can be unknown. However, in real world change point detection, we may not have prior information about how the change will be manifested. For example, when detecting changes in blood glucose, we may know that this variable will change but not precisely the form this will take in terms of how parameters will be affected. The generalized likelihood ratio statistic has high complexity when the number of samples increases..
>
> Neural networks for changepoint detection: We agree that all methods have both advantages and disadvantages. However, we note that our problem is not specifically anomaly detection (which may indeed result in too few samples for training), and on the datasets tested we were able to learn from as few as 4 examples of changes in synthetic data with 8000 timepoints. In the future we aim to reduce the need for supervision and potentially extend our approach to the unsupervised setting entirely.

---

### Meta-Review · Area_Chair1 · 2018-12-15
**Interesting ideas, but insufficient insight and novelty**

**Confidence:** 4
**Recommendation:** Reject

**Metareview:**

This paper studies change-point detection in time series using a multiscale neural network architecture which contains recurrent connections across different time scales.

Reviewers were mixed in this submission. They found the paper generally clear and well-written, and the idea of adding a multiscale component to the model interesting. However, they also pointed out weaknesses in the related work section and found the experimental setup somewhat limited. In particular, the paper provides little to no analysis of the learnt features. Taking these assessments into consideration, the AC concludes this submission cannot be accepted at this time.